# Economic Evaluation of Emotional and Personal Support in the Health Care of Women with Disabilities

**DOI:** 10.3390/healthcare9040438

**Published:** 2021-04-08

**Authors:** Manuel Vargas-Vargas, María-Leticia Meseguer-Santamaría, Francisco Sánchez-Alberola

**Affiliations:** Faculty of Economics and Business Administration, University of Castilla-La Mancha, 02071 Albacete, Spain; mleticia.meseguer@uclm.es (M.-L.M.-S.); francisco.salberola@uclm.es (F.S.-A.)

**Keywords:** disability, economic costs, emotional and personal support, healthcare, women

## Abstract

It is generally accepted that people with disabilities make greater use of health services. Moreover, certain social circumstances alter the intensity of such use. This manuscript seeks to analyze the existing differences in the use of healthcare among women with and without disabilities, to study the impact of emotional and personal support (EPS) on such use and to assess the reduction of the economic cost that this factor entails. Data from the Spanish National Health Survey (SNHS-2017) and updated unit costs of health services have been used to estimate the differences in use attributable to disability and the economic impact of emotional and personal support. The empirical results show an association between disability and perceived EPS, the latter being less common among Spanish women with disabilities. In addition, within this group, EPS significantly influences the levels of use of health services. Finally, the net effect of a perceived EPS increase would translate into a reduction in the economic costs of health care for women with disabilities.

## 1. Introduction

One of the pillars of the European Union is the development of social rights, among which the right of all people to health services appropriate to their needs is emphasized. Along these lines, the Council of Europe Action Plan 2006–2015 for the Promotion of the Rights and Full Participation of People with Disabilities in Society [1], highlights the right of people with disabilities to quality health services. Most recently the European Commission Recommendation of 26 April 2017 established the European Pillar of Social Rights, adopted jointly on 17 November by the European Parliament, the Commission, and the Council of the European Union [2], which expressly states in chapter III, article 16, that “Everyone has the right to timely access to affordable, preventive, and curative health care of good quality”. Therefore, people with disabilities have the right to receive dignified, specialized, and more frequent health services than the rest of the population [3].

Access to health services is defined as “the utilization of services to achieve the best possible health, including potential access and the actual use of such services” [4]. People with disabilities face more barriers to this access than the general population [5,6] and health services are not adapted to their specific characteristics and needs, meaning that they receive a worse service [7].

For women, it has been shown that there are differences in the use of health services and the cost of those services [8,9]. Therefore, it is of interest to study the general determinants of women’s health, not only those related to reproduction [10], which can orientate public health policies. In the case of women with disabilities, this health care is even more inadequate [11], since they need special attention in services such as gynecology, mammography, maternity, family planning, etc. [12,13,14].

In the literature on health economics, the use of health services is explained according to the characteristics of the health system or of individual users, such as socio-economic status, lifestyle, health status or social support [15,16]. Specifically, social support consists of various forms of help, received or perceived, that the person has from his or her social environment, such as emotional help (assistance or affection), personal help (care or services) or informational help (knowledge or skills) [17]. Previous studies have shown that this is correlated with health status [18,19] or the barriers to access to health services [20], with no significant relationship to unmet health needs [15].

The main aim of this study is to assess the effect that emotional and personal support (EPS) has on the use that women with disabilities in Spain make of health services. To this end, the Spanish National Health Survey of 2017 (SNHS-2017) includes a validated version of the Duke-UNC-11 social support questionnaire [21,22], which makes it possible to obtain a synthetic one-dimensional indicator for this variable and to classify the population into groups of high and low perceived emotional and personal support. Likewise, the SNHS-2017 includes information on the Global Activity Limitation Indicator (GALI) [23,24], which allows the operational definition of disability adopted in this study. These two variables allow the analysis, according to [18,19,20] for EPS and to [10,11,12,13,14] for disability, of the first hypothesis of the study: there is a significant inverse association between EPS and disability in the case of Spanish women (H1).

In a second stage, the differences in the use that women with disabilities make of health services are estimated, and then, within this group, the differences in use according to the perceived EPS are estimated. The results allow the analysis of the second hypothesis: a low perceived EPS significantly reduces the use that this group makes of health services (H2).

Finally, in order to add the effect of EPS on the different services and to assess their overall impact, the economic cost of these differences is obtained, constituting an estimate of the economic impact of such support on the health care cost for women with disabilities in Spain. These data allow the analysis of the third and main hypothesis of the study: a high level of EPS can be associated with a net reduction of the economic cost of health care to this group (H3).

## 2. Materials and Methods

This empirical analysis uses microdata from the last Spanish National Health Survey in 2017 (SNHS-2017), available at [25]. It is a statistical operation of the Ministry of Health, Consumer Affairs and Social Welfare, which it carries out in collaboration with the National Statistics Institute of Spain, by virtue of an agreement signed between the two organizations. Data for women have been selected and weighted by factor elevation, representing a total population of 20,050,756 people.

The “International Classification of Functioning, Disability and Health” (ICF) defines disability as a “generic term encompassing impairments, activity limitations, and participation restrictions” [26]. According to the ICF, at the European level, several European institutions have agreed on an operational statistical definition of disability that allows international comparison. It is about considering that persons with disabilities are those “persons who have had limitations in basic daily life activities due to health problems for at least the last 6 months”. This measure, known as the Global Activity Limitation Indicator (GALI), belongs to the Minimum European Health Module (MEHM), and it is present in some of the major European surveys, such as the European Health Interview Survey (EHIS) or the Statistics on Income and Living Conditions (SILC) in OECD studies.

In the Spanish case, the SNHS-2017 also adopts this operational definition of disability; this research considers women with disabilities who have reported that they have limitations, irrespective of whether or not they are severely affected.

The distribution by age, perceived health status, and global activity limitation indicator of Spanish women are shown in Table 1.

To quantify the perceived social support, the SNHS-2017 includes, within its module of health determinants, eleven variables on “emotional and personal support situations in daily life (EPS)”, and a Spanish version, validated and adapted, of the Duke-UNC-11 questionnaire [21]. It is an instrument that measures two dimensions of emotional support: confidential support (having people to communicate with) and affective support (demonstrations of love, affection and empathy) [22]. The eleven items are:EPS-01: Visits with friends and relatives.EPS-02: Help around the house.EPS-03: Praise and recognition for a good job.EPS-04: I have people who care what happens to me.EPS-05: I get love and affection.EPS-06: I get chances to talk to someone about problems at work or with my housework.EPS-07: I get chances to talk to someone I trust about my personal or family problems.EPS-08: I get chances to talk about money matters.EPS-09: I get invitations to go out and do things with other people.EPS-10: I get useful advice about important things in life.EPS-11: I get help when I am sick in bed.

Each item is scored on a Likert Scale from 1 (“Much less than what I want”) to 5 (“As much as I want”). To mitigate the effect of different variances of each item, the score for perceived EPS is obtained by principal component analysis, selecting the first principal component. According to whether their score is below or above average, women have been classified into two groups, “Above average perceived EPS” and “Below average perceived EPS”.

The healthcare module of the SNHS-2017 collects information on the kinds of healthcare services that women have received. For some services, the survey provides the number of times they are used, while for others only whether it has been used or not (dichotomous variables). The selected variables (Si) are:Number of times the respondent has consulted a general practitioner or family doctor in the past four weeks.Number of times the respondent has consulted a specialist in the last four weeks.Number of times respondent has been hospitalized in the past twelve months, excluding childbirth or caesarean sections.Admission to a day hospital over the past twelve months for intervention, treatment, or to have a test done.Use of any emergency service in the past twelve months.Use of other services over the past 12 months: Physiotherapist over the past twelve months (dichotomous variable).Use of other services over the past 12 months: Psychologist, Psychotherapist or Psychiatrist (dichotomous variable).Use of other services over the past 12 months: Nurse or Midwife (dichotomous variable).Diagnostic tests carried out in the past 12 months: X-ray (dichotomous variable).Diagnostic tests carried out in the past 12 months: CAT scan (dichotomous variable).Diagnostic tests carried out in the past 12 months: Ultrasound (dichotomous variable).Diagnostic tests carried out in the past 12 months: MRI (dichotomous variable).

Finally, information about the unit cost of each health service (UCi) has been obtained from several references. In a study on the health costs associated with intimate partner violence in Spain [27], the authors obtain estimates of the monetary unit cost of some services used by women: general practitioner or family doctor; specialist; hospitalizations; emergency service; physiotherapist; psychologist, psychotherapist or psychiatrist; and nurse or midwife. Unit costs for the rest of the health services have been obtained from other sources: admission in a day hospital [28]; X-ray and CAT scan [29]; ultrasound [30]; MRI [31].

Since the costs obtained correspond to different years, all monetary information has been updated to 2019 using the Spanish consumer price index, as shown in Table 2.

## 3. Results

The SNHS-2017 collects eleven variables of emotional and personal support perceived by women, in five-point Likert scales. For all Spanish women, the questionnaire presents a high reliability, with a Cronbach alpha coefficient of 0.899; by disability groups, this reliability is maintained, with alpha coefficients of 0.893 for non-limited women and 0.907 for limited women. Thus, the questionnaire is a valid tool for the study of EPS in the Spanish female population.

Average values of social support, separating limited and non-limited women according to GALI, are shown in Figure 1.

In all the items, perceived EPS is lower among women who have limitations, showing the relationship between such support and the situation of limitation. In general, the first three items have a lower average score for both groups, with the EPS-09 variable reporting the greatest difference between the two groups.

To construct a joint indicator of support, a principal component analysis was performed, with a Kaiser-Meyer-Olkin statistic of 0.922 and a Bartlett’s sphericity test of 140,668,855 (*p*-value < 0.001). A two-dimensional structure has been obtained, where the first principal component, with a 55.268% of explained variance, is associated with the last eight items, related to the affective and confidential subscales of the questionnaire; the second component, with a 10.843% of explained variance, is associated to the first three items, as shown in Table 3.

Since the first principal component is the most related to the support subscales and explains a higher percentage of variance, it has been chosen to be considered as a synthetic indicator of perceived social support. This indicator, once standardized, has an average value of 0.101 for non-limited women in GALI and of −1.181 for limited women, repeating the lower EPS of the second group. Two groups of women have been established following this indicator: those with a negative value (group labeled as “Below average Emotional and Personal Support”) and those with a positive value (group labeled as “Above average Emotional and Personal Support”), whose distribution is shown in Table 4.

As shown, in the case of women with disabilities (limited) the percentage of women who perceive little EPS is significantly higher (40.39%) than among women without disabilities (29.23%). Therefore, there is an inverse association between both variables (Yule’s Q = −0.2427) which is significant (squared chi = 224,852.43, *p*-value < 0.001).

Analyzing the twelve health services selected from those provided by the SNHS-2017, Table 5 shows the average number of times that women use the first five, whereas for the remaining seven, it shows the proportion of women who ever use it. This information is disaggregated according to whether they have reported limitations or not.

For all services, the use is greater among women who declare having limitations, highlighting the admission to a day hospital, with an average of five times more in the case of women with limitations. This increased need for health care makes them a more vulnerable group with a greater impact on the provision of services and, consequently, on health expenditure. Thus, the analysis of social support as a factor that can reduce the use of health services among women with limitations will permit the measurement of its impact on associated expenditure.

In order to quantify the impact of EPS on the use of health services among women with limitations, the averages for the twelve services have been calculated by differentiating between women who receive little and women who receive a lot of EPS. The results are shown in Table 6 next to the number of women in each group.

These values show that there are health services whose use is reduced when women with disabilities perceive greater emotional and personal support, among which are the most widely used services. However, to access more specialized services (visit to specialist, physiotherapist, nurse or midwife and the diagnostic tests X-ray, ultrasound and MRI), women with disabilities may have usage barriers. Therefore, an increased EPS can reduce these barriers and lead to greater use of this type of specialized health services.

Table 6 also permits the estimation of the variation in health use (Δ*U*) that would occur if all women made an average use equivalent to that of the group with more emotional and personal support. For each service (Si), the difference between the average use (*U*) of each group multiplied by the number of women receiving little support would indicate the increase in use (decrease, if negative) that would be achieved if all women belonged to the group of more perceived EPS, as shown in Equation (1):(1)$$\Delta U_{i}=\left(\bar{U_{A_{i}}}-\bar{U_{B_{i}}}\right)N_{B_{i}},$$
where the subindex *A* refers to the group “Above average EPS” and the subindex *B* to the group “Below average EPS”.

The assessment of the economic cost, both of the actual use of each health service (*Ci*) and of the variation associated with EPS (Δ*Ci*), is calculated using the unit costs (*UCi*) shown in Table 2, as shown in Equations (2) and (3):(2)$$C_{i}=\left(\bar{U_{B_{i}}}N_{B_{i}}+\bar{U_{A_{i}}}N_{A_{i}}\right)UC_{i},$$
(3)$$\Delta C_{i}=\left(\bar{U_{A_{i}}}-\bar{U_{B_{i}}}\right)N_{B_{i}}UC_{i},$$

Finally, the sum for all health services provides the total cost of health care and its total variation, as shown in Table 7.

The net balance shows that, if all women with disabilities made an average use of these health services equivalent to that of the group that receives more EPS, there would be a net reduction of 2.49% of the cost of such services, valued at just over €450 million. Especially noteworthy are the economic cost reductions associated with hospitalization and day hospital services, the first due to its higher cost (€ 8906.55, as shown in Table 2) and the second given the higher frequency of use among women with limitations (17.01) compared to women without limitations (3.46), as shown in Table 5.

This net balance includes a decrease in expenditure in six of the twelve services (those where Δ*Ci* < 0), whose aggregate value is €475,687,482.58, which represents 2.63% of the total cost; and an increase in the other six (with Δ*Ci* > 0), valued at €25,367,680.16 (0.14% of total cost). Thus, the increase in EPS perceived by women with limitations would have a double effect: on the one hand, a reduction in the use of some more general health services, valued at a gross reduction in economic costs of 2.63%; and, on the other hand, an increase in the use of more specialized health services, increasing health care for this group, whose cost would be valued only at 0.14%.

## 4. Discussion

Data provided by the Duke-UNC-11 social support questionnaire of the SNHS-2017 show that it is a valid and reliable tool for the Spanish female population in general and for both groups of women with disabilities. These results are equivalent to those obtained for the Spanish population [22,32,33], although with some slight differences in the factorial structure obtained. Similar results are also obtained in the validation for more specific groups, such as the caregiver population [34,35,36] or women who have suffered gender violence in Chile [37].

The eleven partial indicators included in the questionnaire always show a lower value of social support among women with disabilities, as shown in Figure 1. The factorial structure obtained for this questionnaire shows two principal components with eigenvalue greater than the unit. The first component is associated with the items of the factor confidant and affective support, and the second component is associated with the first three items, more neutral, coinciding with the factorial structure obtained in the original study by Broadhead et al. [18]. Thus, this study has chosen to consider as a synthetic indicator of emotional and personal support the score of the first principal component, which collects 55.268% of the original variability and with high factorial loads in the items of the confidant and affective components.

The association between disability and EPS, shown in Table 4, confirms the first hypothesis of this manuscript, showing an inverse and significant relationship between EPS and disability. This result, similar to that obtained in previous studies for quality of life [38], indicates that the group of women with disabilities perceives lower EPS, being more vulnerable to problems related to the lack of emotional and personal support.

Focusing on the use of health services, Table 5 shows that women with disabilities make a greater use of these services. This fact supports the previous evidence of the impact of disability on a worse health state [39,40,41], or on a greater and more specialized need for healthcare [42,43]. Thus, given these special characteristics, a more detailed analysis of the influence of EPS on the use of health services of this group has been addressed.

Table 6 shows the differences in the average use of health services among the groups with below and above average perceived EPS of women with disabilities, constituting a partial determinant of health status [15,44]. In six of the twelve services studied, a higher level of EPS is associated with a reduction in use. These services are the more commonly used ones and focus on a more general health care (general practitioner, hospitalizations, day hospital, emergency services, psychologist, and CAT scans). In this case, EPS improves the conditions which women with disabilities face, so they need less health care.

Moreover, for the remaining six services (visit to specialist, physiotherapist, nurse or midwife and the diagnostic tests X-ray, ultrasound, and MRI) a higher level of EPS is associated with a greater use of these more specialized services. The greatest barriers to access to health services, and the resulting unmet needs of women with disabilities [45], can be partially reduced by personal support, facilitating access to these more specific services.

Thus, both effects show that EPS significantly influences the use women with disabilities make of health services, supporting the second hypothesis of this manuscript.

Finally, Table 7 shows an estimate of the impact of EPS using the unit costs listed in Table 2 to economically value the use of health services (Ci), as well as the variation that would occur if all women had a high level of emotional and personal support (ΔCi). In the six general services stated above, the EPS induces a reduction of the economic cost associated with its reduced use, valued jointly at 2.63% of the cost of these services (€475,687,482.58); most of the reduction in economic cost is concentrated in hospitalization services (mainly due to their higher economic cost, as shown in Table 2) and day hospital services (due to the high frequency of use among women with limitations, as shown in Table 5). For the six more specialized services, a higher level of EPS induces an increase of 0.14% in the monetary cost (€25,367,680.16), much lower than the savings in general services. Not only would there be a net saving of 2.49% in the cost of health care (€450,319,802.43), which supports the third hypothesis of this manuscript, but also the care received by women with disabilities in specialized medical services would be increased, reducing the rate of unmet services. This result can help to direct public policy measures for this particularly vulnerable group towards a better social support outside the health sector.

The results obtained allow a first estimation of the effect that EPS has on the use that women with disabilities make of health services. However, it is necessary to delve into this line of research in order to solve some of the limitations of this study. It should be noted that the statistical information comes from the SNHS-2017, an official survey, not from actual data on the use of health services. The availability of microdata from the health system on its services, the real economic costs disaggregated by provision, and the socio-demographic characterization of users, would allow for more accurate analysis and more useful conclusions for the health administration. Nevertheless, such quantitative information, which is more difficult to obtain, is not available and therefore partial or indirect estimation processes are used [27,28,46,47,48], in this manuscript.

Thus, the calculation of the economic effect of an increase in EPS should be considered as an initial estimate, which should be reviewed as the number of health services included can be increased. More information on the use that patients make or a more detailed assessment of the cost structure can be obtained, aspects which are difficult to implement in a health survey. Likewise, the unit costs of the health services used in the study are estimates for the whole population, without distinction of gender or state of limitation, information that would modify those costs but that is not elaborated. In addition, the variation in cost on the assumption that the entire population studied had a high level of EPS is based on the average use of this subgroup, without delving into the causes of unmet medical needs. Since the SNHS-2017 only includes waiting time on the list, economic reasons and transport problems, this limitation hinders other scenarios.

## 5. Conclusions

In conclusion, despite these limitations, the results shown in this study represent an initial point from which to approach this line of research, since the three hypotheses raised are supported. Firstly, there is an association between disability and perceived emotional and personal support, the latter being less common among Spanish women with disabilities. Secondly, within this group, EPS significantly influences the levels of use of health services. Thirdly, the net effect of a perceived EPS increase would translate into a reduction in the economic costs of health care for women with disabilities.

The association between EPS and use (and cost) of healthcare services has important implications for planning by health service providers, who must be aware of the specific needs of women with disabilities and their differentiated use of these services. Achieving an increase in the emotional and personal support perceived by women with disabilities can induce a reduction in the associated economic cost, concentrated in hospitalization services. The high monetary cost of hospitalization and that related to the greater use of the day hospital service by this group, could be reduced by an estimated value of nearly 423 million euros, which accounts for almost 89% of savings in healthcare spending linked to EPS increase.

Another expected effect of increasing the EPS is related to improving health care for women with disabilities, with an increase in the use of certain specialized services, that could reduce the rate of unmet healthcare services. This result can help to direct public policy measures for this particularly vulnerable group towards a better social support outside the health sector.

## Figures and Tables

**Figure 1 healthcare-09-00438-f001:**
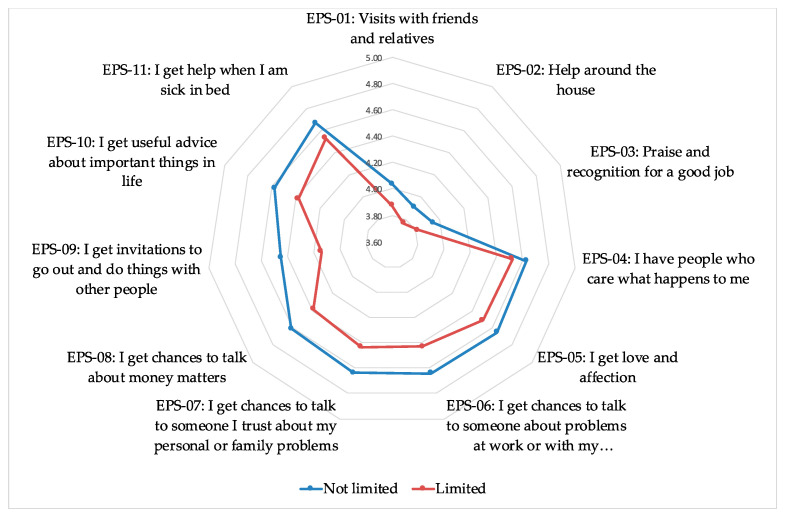
Average of emotional and personal support (EPS) items for limited and not limited women.

**Table 1 healthcare-09-00438-t001:** Sociodemographic characteristics.

**Age**	**Count**	**Frequency**
From 15 to 29 years old	3,371,139	16.81%
From 30 to 49 years old	7,073,835	35.28%
From 50 to 64 years old	4,780,142	23.84%
From 65 to 80 years old	3,359,199	16.75%
More than 80 years old	1,466,441	7.30%
**Perceived Health Status**	**Count**	**Frequency**
Very Good	3,854,482	19.22%
Good	9,470,639	47.23%
Fair	4,739,030	23.64%
Bad	1,514,248	7.55%
Very bad	472,358	2.36%
**Global Activity Limitation Indicator**	**Count**	**Frequency**
Severely limited	1,069,577	5.34%
Limited, but not severely	4,806,974	23.98%
Not limited	14,170,396	70.68%

Source: Own elaboration from SNHS-2017 microdata.

**Table 2 healthcare-09-00438-t002:** Unit costs of healthcare services.

Healthcare Service	Unit Cost (UC_i_), €
Number of times the respondent has consulted a general practitioner or family doctor in the past four weeks	46.98
Number of times the respondent has consulted a specialist in the last four weeks	120.05
Number of times respondent has been hospitalized in the past twelve months, excluding childbirth or caesarean sections	8906.55
Admission to a day hospital over the past twelve months for intervention, treatment, or to have a test done	173.39
Use of any emergency service in the past twelve months	182.68
Use of other services over the past 12 months: Physiotherapist over the past twelve months	24.01
Use of other services over the past 12 months: Psychologist, Psychotherapist or Psychiatrist	100.21
Use of other services over the past 12 months: Nurse or Midwife	24.01
Diagnostic tests carried out in the past 12 months: X-ray	20.70
Diagnostic tests carried out in the past 12 months: CAT scan	37.12
Diagnostic tests carried out in the past 12 months: Ultrasound	26.06
Diagnostic tests carried out in the past 12 months: MRI	112.32

**Table 3 healthcare-09-00438-t003:** Principal components for EPS items.

Emotional and Personal Support Items	First P.C.	Second P.C.
EPS-01: Visits with friends and relatives	0.220	0.744
EPS-02: Help around the house	0.184	0.791
EPS-03: Praise and recognition for a good job	0.249	0.723
EPS-04: I have people who care what happens to me	0.712	0.336
EPS-05: I get love and affection	0.719	0.287
EPS-06: I get chances to talk to someone about problems at work or with my housework	0.893	0.165
EPS-07: I get chances to talk to someone I trust about my personal or family problems	0.903	0.162
EPS-08: I get chances to talk about money matters	0.868	0.177
EPS-09: I get invitations to go out and do things with other people	0.595	0.421
EPS-10: I get useful advice about important things in life	0.757	0.316
EPS-11: I get help when I am sick in bed	0.677	0.308

**Table 4 healthcare-09-00438-t004:** Women by global activity limitation indicator (GALI) and emotional and personal support.

	Global Activity Limitation Indicator			
	Not Limited		Limited	
Emotional and Personal Support	N	%	N	%
Below average EPS	3,994,546	29.23	2,246,794	40.39
Above average EPS	9,672,982	70.77	3,315,579	59.61

**Table 5 healthcare-09-00438-t005:** Use of healthcare services by group of GALI.

Healthcare Service (S_i_)	Not Limited	Limited	F-Statistics (*p*-Value)
Number of times the respondent has consulted a general practitioner or family doctor in the past four weeks	0.36	0.78	1,007,809.37 (<0.001)
Number of times the respondent has consulted a specialist in the last four weeks	0.25	0.51	347,495.36 (<0.001)
Number of times respondent has been hospitalized in the past twelve months, excluding childbirth or caesarean sections	1.17	1.48	31,973.20 (<0.001)
Admission to a day hospital over the past twelve months for intervention, treatment, or to have a test done	3.46	17.01	11,788.97 (<0.001)
Use of any emergency service in the past twelve months	1.69	2.33	67,320.62 (<0.001)
Use of other services over the past 12 months: Physiotherapist over the past twelve months	16.55%	23.61%	137,471.16 (<0.001)
Use of other services over the past 12 months: Psychologist, Psychotherapist or Psychiatrist	4.14%	12.08%	441,789.20 (<0.001)
Use of other services over the past 12 months: Nurse or Midwife	14.92%	26.09%	353,738.17 (<0.001)
Diagnostic tests carried out in the past 12 months: X-ray	20.18%	45.90%	1,471,102.62 (<0.001)
Diagnostic tests carried out in the past 12 months: CAT scan	5.06%	17.75%	870,652.06 (<0.001)
Diagnostic tests carried out in the past 12 months: Ultrasound	19.31%	26.88%	141,287.46 (<0.001)
Diagnostic tests carried out in the past 12 months: MRI	5.58%	17.45%	734,905.14 (<0.001)

**Table 6 healthcare-09-00438-t006:** Average use of healthcare services by women with limitations (EPS groups).

Healthcare Service (S_i_)	Below Average EPS (*N*_B_)	Above Average EPS (*N*_A_)	F-Statistic (*p*-Value)
Number of times the respondent has consulted a general practitioner or family doctor in the past four weeks	0.83 (2,158,411)	0.75 (3,433,199)	5343.93 (<0.001)
Number of times the respondent has consulted a specialist in the last four weeks	0.51 (1,764,468)	0.52 (2,969,236)	646.46 (<0.001)
Number of times respondent has been hospitalized in the past twelve months, excluding childbirth or caesarean sections	1.51 (385,449)	1.45 (622,098)	2452.11 (<0.001)
Admission to a day hospital over the past twelve months for intervention, treatment, or to have a test done	19.48 (329,307)	15.67 (609,365)	151.09 (<0.001)
Use of any emergency service in the past twelve months	2.46 (1,047,554)	2.25 (1,704,129)	955.13 (<0.001)
Use of other services over the past 12 months: Physiotherapist over the past twelve months	20.07% (2,245,468)	25.79% (3,629,757)	27,378.90 (<0.001)
Use of other services over the past 12 months: Psychologist, Psychotherapist or Psychiatrist	13.45% (2,246,794)	11.22% (3,627,255)	8013.95 (<0.001)
Use of other services over the past 12 months: Nurse or Midwife	19.87% (2,246,794)	29.94% (3,621,446)	69,686.00 (<0.001)
Diagnostic tests carried out in the past 12 months: X-ray	45.10% (2,245,385)	46.39% (3,624,818)	1928.72 (<0.001)
Diagnostic tests carried out in the past 12 months: CAT scan	18.05% (2,244,874)	17.57% (3,619,857)	118.09 (<0.001)
Diagnostic tests carried out in the past 12 months: Ultrasound	26.09% (2,241,611)	27.36% (3,629,405)	3185.51 (<0.001)
Diagnostic tests carried out in the past 12 months: MRI	14.32% (2,243,072)	19.39% (3,602,544)	32,314.80 (<0.001)

**Table 7 healthcare-09-00438-t007:** Costs of healthcare services and reduction by EPS.

Healthcare Service (S_i_)	*C_i_* (Million €)	$\Delta C_{i}\;(\mathrm{Million}\;€)$	$\Delta C_{i}/C_{i}$
Number of times the respondent has consulted a general practitioner or family doctor in the past four weeks	205.28	−8.12	−3.95%
Number of times the respondent has consulted a specialist in the last four weeks	292.56	2.73	0.93%
Number of times respondent has been hospitalized in the past twelve months, excluding childbirth or caesarean sections	13,243.09	−205.51	−1.55%
Admission to a day hospital over the past twelve months for intervention, treatment, or to have a test done	2768.44	−217.41	−7.85%
Use of any emergency service in the past twelve months	1170.61	−39.24	−3.35%
Use of other services over the past 12 months: Physiotherapist over the past twelve months	33.30	3.09	9.27%
Use of other services over the past 12 months: Psychologist, Psychotherapist or Psychiatrist	71.09	-5.02	−7.05%
Use of other services over the past 12 months: Nurse or Midwife	36.75	5.43	14.78%
Diagnostic tests carried out in the past 12 months: X-ray	55.77	0.60	1.07%
Diagnostic tests carried out in the past 12 months: CAT scan	38.64	-0.40	−1.04%
Diagnostic tests carried out in the past 12 months: Ultrasound	41.06	0.74	1.80%
Diagnostic tests carried out in the past 12 months: MRI	114.91	12.78	11.12%
ALL HEALTHCARE SERVICES	18,071.49	−450.32	−2.49%

## Data Availability

Not applicable.

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
