# Peer review of "Economic Evaluation of Emotional and Personal Support in the Health Care of Women with Disabilities"

_healthcare, 2021, doi:10.3390/healthcare9040438_

Round 1
Reviewer 1 Report
This manuscript uses good quality data and an interesting combination of instruments. The finding that greater emotional and personal support for disabled women could reduce the utilisation of relatively expensive general treatments is adequately supported by the evidence and has policy relevance. The authors recognise the analysis constitutes only an initial stage in this promising area of research. A few changes to enhance the clarity of the manuscript would further improve this work.
Major points
1. Emotional and personal support is used interchangeably with social support, despite the former not fulfilling the definition of social support given within the introduction. Furthermore, following principal component analysis it seems that personal help aspects of the Duke-UNC-11 have been largely discarded in the measure of social support used within the analysis. Please consider if social support is an appropriate term. Would emotional and personal support, as used within the article title, be better suited throughout the analysis?
2. I would prefer to see a stronger motivation for the focus on women, and to some extent the disabled. Greater detail from the references would suffice.
3. The basis for the three hypotheses should be clearly stated, plus the expected direction of the relationship for H2.
4. The labelling of the social support group “a little” or “much” is misleading. Please consider using ‘above average’ and ‘below average’.
5. Tables 5 and 6 would benefit from having a simple statistical test of means between categories within each healthcare service in a third column. It appears there are fewer differences between EPS groups. The absence of statistically significant differences in service utilisation would impact on subsequent cost estimates. Within the discussion, differences between groups in Table 6 are referred to as “significant”, which may be misleading if differences are not statistically significant.
6. To aid comparison, I would prefer Table 6 and Table 5 to look exactly alike, or even be combined.
7. The lower 7 healthcare services in Table 5 and Tables 6 would be more clearly expressed as a percentage. Please also consider if four decimal places are adding substantial information. One or two decimal places would make the table clearer without substantial loss of information.
8. From Equation 1 onwards, the elements within the formulae are not always clearly defined. For example, the reader must deduce that UM is service utilisation for those with “much EPS”. Maybe I missed where the subscripts were introduced, but it would be helpful to state them again to ensure each equation can be understood without searching earlier in the paper.
9. A great deal of the potential cost savings are concentrated on two health services, both related to hospitalisation. For one of these service in particular, Table 5 shows a much larger difference between the “not limited” and “limited” groups than for any other service. For the other, Table 2 shows a far higher cost than other services. These key drivers of the findings, and potential pathways through which EPS can reduce them, do not currently receive adequate attention within the results and discussion.
Minor points
1. The text within Figure 1 was difficult to read. Could this be made darker?
2. The absolute numbers in Table 6 are poorly formatted.
3. Decimals places in Table 7 are not useful when numbers are so large. They remain useful for percentages. Perhaps the first two columns of results could be expressed as millions of euros.
4. P.2 line 59. Should this be ‘allow us to analyze’ or ‘allow the analysis’. Similar issue on lines 64 and 70. Additionally, line 64 should be ‘perceived social support significantly affects’.
Author Response
Referee 1:
This manuscript uses good quality data and an interesting combination of instruments. The finding that greater emotional and personal support for disabled women could reduce the utilisation of relatively expensive general treatments is adequately supported by the evidence and has policy relevance. The authors recognise the analysis constitutes only an initial stage in this promising area of research. A few changes to enhance the clarity of the manuscript would further improve this work.
Major points
- Emotional and personal support is used interchangeably with social support, despite the former not fulfilling the definition of social support given within the introduction. Furthermore, following principal component analysis it seems that personal help aspects of the Duke-UNC-11 have been largely discarded in the measure of social support used within the analysis. Please consider if social support is an appropriate term. Would emotional and personal support, as used within the article title, be better suited throughout the analysis?
Response: Indeed, Emotional and Personal Support is part of functional social support, measured through the DUKE-UNC-11 questionnaire, and the operational measure used in this manuscript is a linear combination of the 11 items (first main component), although by far less weight of the first three items, more personal; therefore, we have alternated the terms EPS and social support. Following your comment and to be more specific, we will only use the term EPS.
- I would prefer to see a stronger motivation for the focus on women, and to some extent the disabled. Greater detail from the references would suffice.
Response: Following this comment, we have included a new paragraphs in the introduction section (lines 39-42).
- The basis for the three hypotheses should be clearly stated, plus the expected direction of the relationship for H2.
Response: Following this comment, we have modified the writing of the three hypotheses of the manuscript.
- The labelling of the social support group “a little” or “much” is misleading. Please consider using ‘above average’ and ‘below average’.
Response: Following this comment, we have replaced “a little” and “much” with “Above average” and “Below average”.
- Tables 5 and 6 would benefit from having a simple statistical test of means between categories within each healthcare service in a third column. It appears there are fewer differences between EPS groups. The absence of statistically significant differences in service utilisation would impact on subsequent cost estimates. Within the discussion, differences between groups in Table 6 are referred to as “significant”, which may be misleading if differences are not statistically significant.
Response: Following this comment, we have included a new column in tables 5 and 6, with de F-Statistic and p-value for each healthcare service. All differences are statistically significant.
- To aid comparison, I would prefer Table 6 and Table 5 to look exactly alike, or even be combined.
Response: Tables 5 and 6 have almost the same format. The only difference is that Table 6 contains the population sizes of each group (Below and Above average EPS), since they are used later to estimate cost differences; however, population sizes of each group ("Not limited" and "limited") in Table 5 are not used later, so they are not shown to make the table easier to read. We consider it more appropriate to present the results in two different tables, since Table 5 shows results for the total population of women while Table 6 shows results for the group of women with limitations.
- The lower 7 healthcare services in Table 5 and Tables 6 would be more clearly expressed as a percentage. Please also consider if four decimal places are adding substantial information. One or two decimal places would make the table clearer without substantial loss of information.
Response: Following this comment, we have expressed the results in Tables 5 and 6 to two decimal places, and the last seven healthcare services as a percentage.
- From Equation 1 onwards, the elements within the formulae are not always clearly defined. For example, the reader must deduce that UM is service utilisation for those with “much EPS”. Maybe I missed where the subscripts were introduced, but it would be helpful to state them again to ensure each equation can be understood without searching earlier in the paper.
Response: Following this comment, we have included after equation 1 a clarification on the meaning of the subindexes.
- A great deal of the potential cost savings are concentrated on two health services, both related to hospitalisation. For one of these service in particular, Table 5 shows a much larger difference between the “not limited” and “limited” groups than for any other service. For the other, Table 2 shows a far higher cost than other services. These key drivers of the findings, and potential pathways through which EPS can reduce them, do not currently receive adequate attention within the results and discussion.
Response: Following this comment, we have included two explicative paragraphs, one in the results section (lines 510-513) and other in the discussion section (lines 576-578).
Minor points
- The text within Figure 1 was difficult to read. Could this be made darker?
Response: We have replaced Figure 1, increasing the font size and with a darker color.
- The absolute numbers in Table 6 are poorly formatted.
Response: We have reformatted Table 6.
- Decimals places in Table 7 are not useful when numbers are so large. They remain useful for percentages. Perhaps the first two columns of results could be expressed as millions of euros.
Response: We have expressed Table 7 as millions of euros, with two decimals.
- P.2 line 59. Should this be ‘allow us to analyze’ or ‘allow the analysis’. Similar issue on lines 64 and 70. Additionally, line 64 should be ‘perceived social support significantly affects’.
Response: We have corrected lines 59, 64 and 70.

Reviewer 2 Report
This is a good piece of research effort should be made to integrate raw health services use data to see the actual effect. That would be useful for policy purpose and aid in tailoring the needed help for those with any form of disability when accessing healthcare.
Author Response
Referee 2:
This is a good piece of research effort should be made to integrate raw health services use data to see the actual effect. That would be useful for policy purpose and aid in tailoring the needed help for those with any form of disability when accessing healthcare.
Response: Thank you very much for your comments.

Reviewer 3 Report
The paper covers an interesting and important topic.
In overall terms, the paper is well structured and motivated.
The methods are clearly described.
My overall opinion about the paper is positive. I have however three key concerns:
- I feel the lack of some (short) literature review. What is currentely presented in the introduction does not seem enough to provide the adequate background for the empirical analysis;
- the discussion of the results could be stronger, namely through a stronger link with previous evidence on related topics;
- the conclusion is very poor. The author should make an effort to reformulate this section, providing guideleines for policy action.
Author Response
Referee 3:
The paper covers an interesting and important topic.
In overall terms, the paper is well structured and motivated.
The methods are clearly described.
My overall opinion about the paper is positive. I have however three key concerns:
I feel the lack of some (short) literature review. What is currentely presented in the introduction does not seem enough to provide the adequate background for the empirical analysis;
Response: We have expanded the introductory section, including some references, and reformulated hypotheses to clarify their aims.
the discussion of the results could be stronger, namely through a stronger link with previous evidence on related topics;
Response: We have introduced some changes in the discussion section. The influence of the EPS on the use of health services and their cost in the group of women with disabilities is a subject little discussed in the literature, making it difficult to compare our results with previous ones. This difficulty is listed as a limitation, and other approaches used in the literature to estimate the health costs are presented.
the conclusion is very poor. The author should make an effort to reformulate this section, providing guideleines for policy action.
Response: Following this comment, we have reformulated the conclusions section, adding some guidelines for policy action.

Round 2
Reviewer 3 Report
The new version of the paper answers to my previous concerns. Thus, in my opinion, the paper can be accepted for publication.